# Genomics knowledge and attitudes among European public health professionals: Results of a cross-sectional survey

Annalisa Rosso[1,2]*, Erica Pitini[1], Elvira D'Andrea[1,3], Marco Di Marco[1,4], Brigid Unim[1], Valentina Baccolini[1], Corrado De Vito[1], Carolina Marzuillo[1], Floris Barnhoorn[5], Dineke Zeegers Paget[5], Paolo Villari[1]

1 Department of Public Health and Infectious Diseases, Sapienza University of Rome, Rome, Italy, 2 Local Health Unit-Azienda Sanitaria Locale Roma 2, Rome, Italy, 3 Division of Pharmacoepidemiology and Pharmacoeconomics, Department of Medicine, Brigham and Women's Hospital and Harvard Medical School, Boston, MA, United States of America, 4 Local Health Unit Azienda Sanitaria Locale Roma 1, Rome, Italy, 5 European Public Health Association (EUPHA), Utrecht, the Netherlands

* annalisa.rosso@uniroma1.it

## Abstract

**Data Availability Statement:** All relevant data are within the manuscript and its Supporting Information files.

### Background

The international public health (PH) community is debating the opportunity to incorporate genomic technologies into PH practice. A survey was conducted to assess attitudes of the European Public Health Association (EUPHA) members towards their role in the implementation of public health genomics (PHG), and their knowledge and attitudes towards genetic testing and the delivery of genetic services.

### Methods

EUPHA members were invited via monthly newsletter and e-mail to take part in an online survey from February 2017 to January 2018. A descriptive analysis of knowledge and attitudes was conducted, along with a univariate and multivariate analysis of their determinants.

### Results

Five hundred and two people completed the questionnaire, 17.9% were involved in PHG activities. Only 28.9% correctly identified all medical conditions for which there is (or not) evidence for implementing genetic testing; over 60% thought that investing in genomics may divert economic resources from social and environmental determinants of health. The majority agreed that PH professionals may play different roles in incorporating genomics into their activities. Better knowledge was associated with positive attitudes towards the use of genetic testing and the delivery of genetic services in PH (OR = 1.48; 95% CI 1.01–2.18).

**Funding:** The study was conducted within the project "Personalized pREvention of Chronic DIseases consortium (PRECeDI)" funded by the European Union Horizon 2020 research and innovation programme MSCA-RISE-2014 (Marie Skłodowska-Curie Research and Innovation Staff Exchange), under grant agreement N˚645740.

**Competing interests:** The authors have declared that no competing interests exist.

## Conclusions

Our study revealed quite positive attitudes, but also a need to increase awareness on genomics among European PH professionals. Those directly involved in PHG activities tend to have a more positive attitude and better knowledge; however, gaps are also evident in this group, suggesting the need to harmonize practice and encourage greater exchange of knowledge among professionals.

## 1. Introduction

The development of precision medicine (PM, also referred to as "personalized medicine") has generated a debate within the international public health (PH) community on the opportunity to use genetic information and genomic applications in preventive medicine. However, the introduction of PM interventions into PH practice is still controversial. On one hand, the more skeptical have argued that investing in genomics may divert resources away from basic public health services and from addressing the structural causes of ill health [1–4]. On the other hand, public health professionals and associations worldwide have stressed that the potential of PM to identify high risk individuals and develop tailored preventive interventions cannot be ignored [5–8].

Over recent decades, several professionals have supported the incorporation of genome-based knowledge and technologies into PH, leading to the emergence of public health genomics (PHG) as a multidisciplinary field promoting the appropriate translation of genomics research into health benefits for individuals and populations [9,10]. PH professionals may play different roles in this translation, from evaluating the effectiveness and cost-effectiveness of genomic applications to modelling and assessing the implementation of evidence-based genomic applications into medical practice [9]. One of the priorities of the PHG movement is to achieve adequate knowledge and capacity among PH professionals to facilitate the integration of genomic information into PH activities [11–15]. What is the current level of competence of PH professionals in genomics? Several surveys have been performed to evaluate the knowledge, attitudes and professional behavior of physicians towards the integration of human genomic discoveries into clinical practice [16–21], but only one study targeted PH practitioners [22] and it focused solely on predictive genetic tests for chronic diseases. We conducted a survey on a sample of European PH professionals belonging to the network of the European Public Health Association (EUPHA), which represents European PH professionals, to assess their attitudes towards their role in the implementation of PHG, and their knowledge and attitudes regarding genetic testing and genetic services.

## 2. Material and methods

The study was conducted within the EU-funded project PRECeDI (Personalized PREvention of Chronic DIseases).

A specific questionnaire was developed for the survey, consisting of 33 items grouped into five sections (see S1 Text):

A. Professional details (four questions);

B. Professional activity (seven questions);

C. Knowledge of genetic testing and delivery of genetic services (four questions);

D. Attitudes towards genetic testing and delivery of genetic services (four questions);

E. Attitudes towards the role of PH professionals in PHG (six questions).

"The development of the questionnaire was informed by a literature review and by similar studies previously carried out by the same research group [18, 22]. It was also based on the inputs received by the members of EUPHA Section on PHG and by all partners of PRECeDI project, as described elsewhere [23]

"For the purpose of the survey, we defined genetic testing as "performing a type of medical test involving an analysis of human chromosomes, DNA, RNA, genes, and/or gene products (e.g., enzymes and other types of proteins), which is predominately used to detect heritable or somatic mutations, genotypes, or phenotypes related to disease and health", following the definition proposed by Burke [24]. Further to discussion with other PRECEDI project partners, we have decided to adopt the classification of genetic testing used by the European Society of Human Genetics [25]. In particular, the survey was aimed at investigating knowledge and attitudes towards the implementation of susceptibility (also known as predisposition) tests, defined as "the detection of genetic variants that are associated with an increased risk of disease but cannot predict with certainty the development of disease, because of the incomplete penetrance of the genetic mutation". Examples of this type of tests include BRCA testing, Lynch syndrome testing and familial hypercholesterolemia genetics tests. A "genetic service" was defined as "the provisions to diagnose, advise and treat individuals with risk factors for genetic disorders." According to our interpretation, the delivery of a genetic service is therefore expected to provide not only genetic testing and counselling, but also treatment and follow up of individuals with genetic disorders, similarly to what was previously described by Battista et al.[26]. Statements included in section E were based on published literature dealing with the relationship between public health and human genomics and the possible role of public health professionals in putting PHG into practice [27] A filter question included in section B gave access to an extended version of the questionnaire, including four additional items in both sections C and D, for professionals who answered that they were involved in PHG activities. A pilot phase of the survey was conducted on 61 staff members of the Department of Public Health and Infectious Diseases of Sapienza University in Rome and 10 members of the Department of Genetics from the Vrije University in Amsterdam, to test clarity of language, practicability and interpretation of answers. Internal consistency was also assessed by obtaining Cronbach's alpha coefficients. The questionnaire proved to be reliable in assessing attitudes towards the role of PH professionals in PHG, and it was slightly revised before distribution to the EUPHA network to improve the quality of some questions [23].

The questionnaire was administered online from March 2017 to February 2018. An invitation to participate in the survey was included in the EUPHA monthly newsletter of February 2017; furthermore, the Presidents of five EUPHA sections (Public Health Genomics, Public Health Epidemiology, Public Health Monitoring and Reporting, Public Health Practice and Policy, Chronic Diseases) sent an invitation to their members to respond to the survey. A final reminder was sent to all EUPHA members in September 2017. The first page of the survey included a description of its objective and of the different section of the questionnaire, and informed participants that all data collected were anonymous. Thus, no consent was requested to participate in the survey. The Ethical Committee of Sapienza University of Rome granted its approval to conduct the survey.

Survey responses were collected in an electronic spreadsheet (see S1 Dataset). A descriptive analysis was performed to define the distribution of socio-demographic characteristics of the sample and to assess rates of positive/negative attitudes towards and knowledge of PHG (frequencies, percentages, mean values and SD were calculated). An analysis of determinants of knowledge and attitudes on genetic testing and the delivery of genetic services, and on the

roles of PH professionals in incorporating genomics into PH activities was conducted through the construction of multiple logistic regression models. The variables "knowledge on PHG", "attitudes on PHG" and "attitudes on the role of PH professionals" originally consisting of multiple categories, were collapsed into two levels, adapting the methodology previously used in other surveys conducted by our research group [18, 22, 28]: a *high level of knowledge* of PHG was attributed to respondents providing correct responses to at least three of four questions included in section C of the questionnaire; *positive attitudes* towards PHG were defined as positive attitudes towards the three statements included in section D of questionnaire, except for question D1, which was excluded because it showed a totally different trend in the univariate analysis; for "attitudes on the role of PH professionals", responders with *positive attitudes* were those who agreed with all six statements included in section E. Covariates included in the models were: group of professionals (involved in PHG activities vs not involved), gender, age, history of genetic conditions, exposure to information on genetic testing during undergraduate or post-graduate education, area of degree (with medicine as reference category), sector of work (with the academic sector chosen as reference), main area of work.

Multiple logistic regression models were built using the strategy suggested by Hosmer and Lemeshow [29]. Each variable was examined by univariate analysis using the appropriate statistical test (Student's t-test or $\chi^2$ test) and was included in the model when the p-value was less than 0.25. Subsequently, multivariate logistic regression with backward elimination of any variable that did not contribute to the model on the grounds of the Likelihood Ratio test (cut-off, p = 0.05) was performed. Adjusted odds ratios (ORs) and 95% confidence intervals (CIs) were calculated. All statistical calculations were performed using Stata version 15.0 (Stata Corporation, College Station, TX, USA).

Approval to conduct the study was granted from the Ethics Committee of Sapienza University of Rome.

## 3. Results

Six hundred and twenty-two people accessed the survey and 502 completed all sections (80.7%). Respondents came from all EU28 Countries and some non–EU Countries, including Albania (n = 1), Andorra (n = 1), Bosnia and Herzegovina (n = 4), Norway (n = 15), Russia (n = 1), Serbia (n = 3), Switzerland (n = 25), Turkey (n = 7), and there were 27 people working in developing countries at the time the survey was conducted The response rate obtained was approximately 10%. Table 1 summarizes the main socio-demographic characteristics of the respondents. There were no significant differences in the socio-demographic characteristics between people who completed the whole survey and those who did not, except for the area of degree, with medical doctors being less likely to complete it (p = 0.026).

For knowledge of genetic testing, only 1.7% of respondents (9/526) correctly identified all applications of genetic testing that are based (or not based) on evidence of effectiveness, while 26.1% correctly identified at least eight out of 10 (141/526), which we set as the threshold for a correct response to this question (Table 2). Nearly 30% of respondents correctly identified all clinical conditions for which there is (or is not) strong evidence supporting the use of a genetic test (CDC tier-1) (Table 2). When asked about the professionals involved in the delivery of genetic testing, nearly 50% of respondents (49.4%, 256/518) correctly indicated that such professionals could equally be general practitioners or geneticists or oncologists (Table 2). Most respondents (91.0%, 472/519) knew that predictive/predisposition genetic tests must be associated with genetic counseling (Table 2).

**Table 1. Socio-demographic characteristics of respondents.**

| CHARACTERISTICS | N | (%) |
|---|---|---|
| *Gender (n = 593)** | | |
| Female | 287 | 48.4 |
| Male | 306 | 51.6 |
| *Age (n = 591)** | | |
| 25–40 | 197 | 33.3 |
| 41–55 | 219 | 37.1 |
| 56–75 | 175 | 29.6 |
| *Type of health professional (n = 576)** | | |
| PH professional not involved in PHG | 440 | 76.4 |
| PH professional involved in PHG | 57 | 9.9 |
| Not PH professional and not involved in PHG | 27 | 4.7 |
| Not PH professional but involved in PHG | 52 | 9.0 |
| *Area of degree (n = 574)** | | |
| Medicine | 295 | 51.4 |
| Health professions (e.g nursing) | 50 | 8.7 |
| Biology | 30 | 5.2 |
| Public health | 73 | 12.7 |
| Other (e.g. statistics, political sciences) | 126 | 21.9 |
| *Sector of work (n = 573)** | | |
| Academic | 337 | 58.8 |
| Hospital | 28 | 4.9 |
| Government (national or local) | 113 | 19.7 |
| Public health service | 33 | 5.8 |
| Other (e.g. NGO, technical agency) | 62 | 10.8 |
| *Information on genetic screening in undergraduate training (n = 573)** | | |
| Yes | 242 | 57.8 |
| No | 331 | 42.2 |
| *Information on genetic screening in postgraduate training (n = 574)** | | |
| Yes | 245 | 42.7 |
| No | 278 | 48.4 |
| Not applicable | 51 | 8.9 |

*Number of respondents to the question.

In the subgroup analysis on PH professionals declaring an involvement in PHG activities, only 12.1% of them (4/33) correctly indicated the number of clinical conditions with available evidence supporting the implementation of a genetic test to predict disease risk, while 32.1% (15/53) correctly identified all components of a genetic service; in particular, less than half of respondents thought that treatment, follow-up and clinical surveillance can also be considered within the components of a genetic service (respectively 37.7%, 49.1%, and 47.2%) (see Tables 2_bis in S1).

Most respondents showed positive attitudes towards the implementation of genetic testing and the delivery of genetic services, except for the idea that it would be more important to invest resources in the social and environmental causes of ill health than in the implementation of genetic testing, with over 60% of the sample agreeing with this statement (Table 3).

Positive attitudes were also reported in the extended version addressed to professionals involved in PHG, ranging from 77.6% of respondents (38/49) indicating that the application of

**Table 2. Knowledge of genetic testing and the delivery of genetic services, n (%).**

| Which of the following applications of genetic testing are based on evidence of effectiveness? (multiple answers are possible) (n = 526)*[a] | No | Yes |
|---|---|---|
| Diagnose disease | 189 (35.9) | **337 (64.1)** |
| Determine the severity of a disease | **420 (79.8)** | 106 (20.2) |
| Identify genetic mutations that are responsible for an already diagnosed disease | 209 (39.7) | **317 (60.3)** |
| Identify genetic mutations that may increase the risk of developing a disease | 139 (26.4) | **387 (73.6)** |
| Identify genetic mutations that could be passed on to children | 163 (31.0) | **363 (69.0)** |
| Identify genetic mutations that influence the process of ageing | **427 (81.2)** | 99 (18.8) |
| Guide doctors in deciding on the best treatment to use for certain individuals | 276 (52.5) | **250 (47.5)** |
| Guide doctors in designing an optimal individualized weight loss diet | **489 (93.0)** | 37 (7.0) |
| Ascertain the gender of a fetus | **354 (67.3)** | 172 (32.7) |
| Screen newborn babies for certain treatable conditions | 187 (35.6) | **339 (64.4)** |
| *For which of the following clinical conditions is there currently a base of synthesized evidence supporting the implementation of genetic testing to predict disease risk? (multiple answers are possible) (n = 525)*[b] | No | Yes |
| Hereditary ovarian cancer | 138 (26.3) | **387 (73.7)** |
| Lynch syndrome (hereditary nonpolyposis colorectal cancer) | 231 (44.0) | **294 (56.0)** |
| Gastric cancer | **461 (87.8)** | 65 (12.2) |
| Metastatic non-small-cell lung cancer | **470 (89.5)** | 55 (10.5) |
| Prostate cancer | **452 (86.1)** | 73 (13.9) |
| Alzheimer's disease | **370 (70.5)** | 155 (29.5) |
| Familial hypercholesterolemia | 264 (50.3) | **261 (49.7)** |
| Type 2 diabetes | **413 (78.7)** | 112 (21.3) |
| Acute myeloid leukemia | **447 (85.1)** | 78 (14.9) |
| Depression | **487 (92.8)** | 38 (7.2) |
| *Which of the following professionals may be involved in the delivery of genetic testing?(n = 509)** | | |
| A. General practitioner | | 9 (1.7) |
| B. Geneticist | | 61 (11.8) |
| C. Oncologist | | 9 (1.7) |
| D. All of the above | | **256 (49.4)** |
| E. B+C | | 183 (35.3) |

(*Continued*)

**Table 2.** (Continued)

| Which of the following applications of genetic testing are based on evidence of effectiveness? (multiple answers are possible) (n = 526)*ᵃ | No | Yes |
|---|---|---|
| *Performing susceptibility (or predisposition) tests should necessarily be associated with genetic counseling that includes information, informed consent, and discussion of the results (n = 519)** | | |
| Agree | | **472 (91.0)** |
| Uncertain | | 35 (6.7) |
| Disagree | | 12 (2.3) |

*Number of respondents to the question.

ᵃA correct response to the question was defined as having correctly identified at least 8/10 applications of genetic testing that are based (or not) on evidence of effectiveness.

ᵇA correct response to the question was defined as having correctly identified all conditions for which there is (or is not) currently evidence supporting the implementation of genetic testing.

Percentages referring to correct answers are in bold.

genetic testing in healthy family members of individuals with hereditary chronic diseases may increase prevention opportunities to 98.0% (48/49) agreeing (or strongly agreeing) that specific training initiatives are needed for PH professionals to develop their capacity to evaluate the quality of genetic services (see Tables 3_bis in S1).

There were high rates of agreement with the proposed roles that PH professionals may play in putting PHG into practice (Table 3). In particular, percentages of agreement ranged from 88.4% of respondents (448/507) who agreed (or strongly agreed) that public health thinking should consider that risk factors can affect subsets of the population differently based on genetic susceptibility to 78.7% of respondents (399/507) who agreed (or strongly agreed) that public health programs should actively implement genomic applications that are evidence-based (e.g. *BRCA* testing for relatives of known mutation carriers) (Table 3).

Table 4 summarizes the results of the multivariate analysis. A *high level of knowledge* was associated with indicating PHG as one of the main areas of work, having graduated in medicine, having received training on genetic testing during undergraduate education and working in a PH service. A high level of knowledge of genetic testing and genetic services was associated with *positive attitudes* towards PHG, while academics were less likely to be positive on the topic. People working directly in PHG were more likely to disagree with the importance of investing resources in social and environmental causes of health rather than in genomics. Finally, *positive attitudes* towards the role of PH professionals in implementing PHG were associated with working in PHG and with having received information on PHG during undergraduate training.

## 4. Discussion

This survey revealed poor knowledge of genetic testing and the delivery of genetic services in a representative sample of European PH professionals. However, overall, attitudes towards both the use of genetic testing and delivery of genetic services, and the involvement of PH professionals in putting PHG into practice, were positive. The only negative sentiment was in response to the proposal that it was more important to invest resources in the social and environmental causes of ill health than in the implementation of genetic testing. Over 60% of respondents agreed in fact that public health resources should be targeted mainly at addressing the structural causes of ill health.

**Table 3. Attitudes towards genetic testing and the delivery of genetic services, and towards the role of PH professionals in PHG (% of answers).**

| Statement | Strongly agree | Agree | Neither agree nor disagree | Disagree | Strongly disagree |
|---|---|---|---|---|---|
| *It is more important to invest resources in the social and environmental causes of ill health than in the implementation of genetic testing (n = 522)** | 27.6% | 32.8% | 25.3% | **13.2%** | **1.1%** |
| *Susceptibility (or predisposition) tests should be introduced in the clinical and public health practice even without health interventions with proven efficacy (n = 522)** | 3.0% | 12.8% | 10.0% | **41.6%** | **32.6%** |
| *Susceptibility (or predisposition) tests should be introduced in the clinical and public health practice only if economic evaluations show cost-effectiveness ratios favorable compared with alternative health interventions (n = 522)** | **14.2%** | **44.5%** | 18.8% | 18.8% | 3.6% |
| *Genetic tests for diseases that could have a fatal outcome (e.g. BRCA testing for breast and ovarian cancer) should be provided free at the point of delivery to people who could benefit from them (n = 522)** | **28.3%** | **43.7%** | 11.5% | 12.3% | 4.2% |
| *Public health thinking should consider that risk factors can affect subsets of the population differently based on genetic susceptibility. (n = 507)** | **26.6%** | **61.7%** | 6.9% | 3.8% | 1.0% |
| *Public health professionals should be involved in the continuous assessment of the utility and validity of emerging genomic applications (n = 507)** | **37.6%** | **52.3%** | 8.1% | 1.8% | 0.2% |
| *Public health programs should actively implement genomic applications that are evidence-based (e.g. BRCA testing for relatives of known mutation carriers). (n = 507)** | **27.2%** | **51.5%** | 17.4% | 3.7% | 0.2% |
| *Public health professionals should measure the utilization of genetic services in order to assess unmet needs and inequalities of access to services (n = 507)** | **29.9%** | **50.7%** | 14.8% | 2.8% | 1.8% |
| *Public health professionals should measure in practice outcomes, process indicators and value added of genomic applications (n = 507)** | **34.4%** | **49.2%** | 13.8% | 1.8% | 0.8% |
| *I think that in the future public health programmes (e.g. cancer screening, chronic diseases prevention programmes) will make a greater use of genetic information. (n = 507)** | **29.8%** | **55.6%** | 12.6% | 1.8% | 0.2% |

*Number of respondents to the question.

Percentages referring to positive attitudes towards genetic testing and delivery of genetic services, and towards the proposed roles of PH professionals are in bold.

Nevertheless, over 70% of participants agreed that genetic tests for diseases that could have a fatal outcome (e.g. *BRCA* testing for breast and ovarian cancer) should be provided free at the point of delivery for those who could benefit from them. High rates of respondents also thought that, in the future, PH programmes (e.g. cancer screening, chronic disease prevention programmes) will make greater use of genetic information. Therefore, PH professionals seem to agree in principle with the importance of including genomic applications in PH practice, but at the same time they seem to fear that this could divert resources from addressing the "traditional" determinants of health. There is a long-standing debate in the PH community on the opportunity to focus on -omics and personalized medicine, and whether this may divert from the traditional population-based approach. However, the need to go beyond the dichotomous high-risk versus population approach was suggested, taking into account that also precision public health could contribute to improving population health and achieving social justice—equity, social inclusion, and empowerment [5].

Given the cost constraints currently faced by European healthcare systems, some prioritization criteria are needed to decide which genetic services should be funded from public budgets [30], and these should include consideration of medical benefit, health needs and costs. In this regard, a high proportion of respondents agreed with the idea that evidence on effectiveness and cost-effectiveness should guide the decision of whether to introduce genetic and genomic applications into clinical and PH practice. Such evidence is already robust for some genetic tests of chronic clinical conditions, such as hereditary breast and ovarian cancer, Lynch syndrome and familial hypercholesterolemia [31–34]. However, according to the responses recorded in our survey, many PH professionals are not fully aware of this: 26% for hereditary breast/ovarian cancer, 44% for Lynch Syndrome and 50% for familial hypercholesterolemia

**Table 4. Multivariate analysis of determinants of knowledge and attitudes (only significant results shown).**

| Variables | OR | 95% CI | P value |
|---|---|---|---|
| *Model 1*: Higher level of knowledge of genetic testing and the delivery of genetic services[a] | | | |
| PHG as main area of work (0 = no, 1 = yes) | 6.35 | 2.62–15.33 | 0.000 |
| Information on PHG during undergraduate training (0 = no, 1 = yes) | 1.53 | 1.02–2.30 | 0.040 |
| Area of degree (0 = other, 1 = medicine) | 1.77 | 1.16–2.69 | 0.007 |
| Sector of work (0 = other, 1 = public health service) | 1.98 | 1.05–3.73 | 0.000 |
| *Model 2*: Positive attitudes towards genetic testing and delivery of genetic services[b] | | | |
| Sector of work (0 = other, 1 = Academic) | 0.67 | 0.47–0.97 | 0.032 |
| Knowledge (0 = score≤ 2, 1 = score 3, 4 | 1.48 | 1.01–2.18 | 0.048 |
| *Model 3*: Positive attitudes towards the use of resources for genetic testing[c] | | | |
| PHG as main area of work (0 = no, 1 = yes) | 9.10 | 3.95–20.94 | 0.000 |
| *Model 4*: Positive attitudes towards the role of PH professionals in PHG[d] | | | |
| PHG main area of work (0 = no, 1 = yes) | 4.04 | 1.48–11.02 | 0.006 |
| Information on genetic testing during undergraduate training (0 = no, 1 = yes) | 1.74 | 1.21–2.48 | 0.003 |

[a] Respondents were classified as those who answered correctly to three out of four questions addressing knowledge of genetic testing and the delivery of genetic services (Table 2) vs. all others.

[b] Respondents were classified as those who showed a positive attitude towards all the statements addressing attitudes towards genetic testing (Table 3) except the first one, which was analyzed separately.

[c] Respondents were divided into those who declared they disagreed or strongly disagreed with the statement "*It is more important to invest resources in the social and environmental causes of ill health than in the implementation of genetic testing*"(value = 1) vs all others (value = 0).

[d] Respondents were classified as those who showed a positive attitude towards all the statements addressing the possible role of PH professionals in implementing PHG vs all others.

did not know that the evidence supported the implementation of the relative genetic testing to predict disease risk for these conditions. More widespread knowledge of those genetic and genomic applications with documented evidence of effectiveness and cost-effectiveness would therefore probably improve attitudes towards both the inclusion of PM in preventive interventions and the funding of such interventions through public budgets.

Another result that supports the thesis of improving knowledge to improve attitudes is the association between agreement with investing resources in genomics and involvement in work activities dealing with personalized medicine. This finding highlights the importance of increasing awareness of the opportunities that PM can offer, particularly among PH professionals not directly involved in genomics projects and/or activities. Communication and training needs have been also identified as key challenges in the implementation of PM by the European PerMed Consortium [35], while the PRECeDI consortiuum added among its recommendations the training of. of clinical and public health professionals should be promoted with the aim of reducing inappropriate use in healthcare [36].

Although professionals involved in PHG activities are better informed about genetic testing, gaps in knowledge were also identified in this group. This may be due to differences in recommendations for use of genetic testing, to a lack of standard procedures for the evaluation of genetic services, and to differences in how genetic services are delivered in different countries. It has in fact been reported that, internationally, different frameworks are adopted for the evaluation of genetic testing [37]. Furthermore, organizations that develop guidelines on the implementation of genomic medicine show wide variation in the use of external and systematic reviews, as well as in the updating of recommendations, when they are assessing the strength of relevant scientific evidence [38]. At the European level, it has been emphasised that

guidelines and policies to support the integration of genomics policies into existing healthcare systems need to be harmonized across countries [26,39]. It has also been highlighted that current genetic services are usually delivered without an internationally standardized set of process and outcome measures, making the evaluation of healthcare services difficult [40]. Only a limited number of PHG professionals considered treatment, follow-up and clinical surveillance to be a component of a genetic service. This may be due to differences in the organizational models of genetic services across countries: research on this topic has in fact shown that the definition of a genetic service may vary largely across different settings [41–44].

There was strong agreement that PH professionals should foster the integration of genomics into PH practices, and over 80% of respondents agreed that in the future PH programs will make greater use of genetic information. Positive attitudes towards the involvement of PH professionals in genomics were associated with being trained in genetic testing during undergraduate education and with direct involvement in PHG activities.

It seems clear from these findings, therefore, that greater knowledge of genomics and personalized medicine among health professionals not directly involved in genetics would improve the capacity of health systems to incorporate new genomic technologies [11–14, 35]. Increasing PH capability in genomics could also help to avoid uncontrolled implementation of technologies without proven benefits, which can lead to inappropriate management of patients, detrimental effects on patient health, and waste of financial resources [36]. Our results are in line with the findings of two surveys on the genomics knowledge and attitudes of PH educators conducted respectively in Italy and the US [22,45], showing that knowledge is a significant predictor of positive attitudes towards the use of genetic testing and the delivery of genetic services in PH.

Our findings suggest that healthcare professional education could be the first step to increase knowledge and capacity of professionals on genomics. There are significant differences in the way in which professional education is delivered across the countries of Europe and also the study curricula of physicians and non physician PH specialists may differ substantially. A set of core competencies in genomics for health professionals (even though not specifically addressed at PH professionals) has been proposed by the European Society of Human Genetics, to provide an appropriate framework for establishing minimum standards of preparation and guide the development of study curricula for health-care professionals in all Countries (graduate or post- graduate) [45]. Based on this framework, a study conducted in Italy tried to identify a set of core competencies in genetics for non-geneticists, both physicians and non-physicians, developing a proposal of three different curricula according to the profession, including basic knowledge, but also attitudes and abilities needed to be able to effectively incorporate genomics into practice. [46]. Given the association observed between a direct involvement in PHG activities and both knowledge and attitudes, it may be useful to develop specific training initiatives for the PH workforce in framework of continuous medical education and/or on-job training,

This is the first survey conducted at the European level on knowledge and attitudes of PH professionals relating to genomics, and thus provides useful insights on the topic. However, despite using several measures to maximise it, the response rate was low and therefore the results only apply to approximately 10% of the whole EUPHA membership. The low response rate reflects one of the main challenges of web-based surveys and is coherent with other studies that also relied on this type of tool. Based on the results of different studies, Ban Mol reported in fact that a response rate below 10% is not uncommon for web surveys, which have been shown to generally get a 6 to 15% lower response rate compared to other survey modes [47]. The low response rate in our study was also influenced by our choice not to opt for an "aggressive" approach to reminders: no telephone calls were used, as in our previous surveys, and we chose to only sent a limited number of reminders (three emails were sent in total to EUPHA

members), considering that repeated follow-ups have been shown to diminish returns and may have the counterproductive effect to irritate potential respondents, without noticeably increasing response rates [48].

A selection bias may therefore have occurred, with respondents differing systematically from non-respondents. The sample proved to be representative of the population of EUPHA members in terms of professional background: a recent survey conducted by the EUPHA secretariat indicated that 66% of responders identified themselves as researchers, 14% as policymakers and the remaining 20% as practitioners (data not published), showing a similar distribution than our sample. However, due to the anonymous nature of the survey, it was not possible to assess any other difference between responders and non responders. We can assume that people responding to the survey had, in general, a stronger interest in genomics than non-responders, and therefore showed more positive attitudes. Given the lower level of knowledge of genomics among people not involved in this area of work, we can also assume that genomics knowledge among the European PH community is even lower than reported in our study, reinforcing the apparent need to increase the genomics capability of professionals not directly dealing with this area. The fact that our findings are consistent with those of studies conducted in the PH community in Italy in 2010 and among US public health educators in 2008 [22, 49], and that responses were obtained from all EU countries, suggests that our conclusions are generally applicable across European and other nations. Another limitation of the study was the impossibility to compare knowledge and attitudes on genomics across different Countries: respondents came from a high number of Countries (over 35), not allowing to stratify responses according to the provenience. The lack of information on the organizational models of genetic services in all Countries did not allow to control for type of delivery model either [50].

In conclusion, our sample of PH professionals in Europe was positive about incorporating the increasing number of genomic applications into their working practice, and agreed they should play a role in this translation process, but their knowledge of PHG is rather weak. Specific efforts should be made to increase the capacity of PH professionals not directly involved in genomics, by implementing communication and/or training strategies at national and international levels. Our findings also highlight an urgent need to further develop and share common and standardized definitions and operational guidelines on PM within the PH community in Europe. In this respect, joint research and training initiatives on PM and genomics at the European level, such as the PRECeDI consortium, have proved to be useful in developing and sharing evidence in this field and should continue to be promoted.

## Supporting information

**S1 Text. Questionnaire.**
(DOC)

**S1 Dataset. Survey data.**
(XLS)

**S1 Table. Table 2_bis and Table 3_bis.**
(DOCX)

## Acknowledgments

The authors are grateful to the European Public Health Association (EUPHA) Office for the support received in conducting the survey.

## Author Contributions

**Conceptualization:** Annalisa Rosso, Erica Pitini, Elvira D'Andrea, Marco Di Marco, Brigid Unim, Valentina Baccolini, Corrado De Vito, Carolina Marzuillo.

**Data curation:** Annalisa Rosso, Valentina Baccolini, Carolina Marzuillo.

**Formal analysis:** Annalisa Rosso.

**Investigation:** Annalisa Rosso.

**Methodology:** Annalisa Rosso, Erica Pitini, Elvira D'Andrea, Marco Di Marco, Brigid Unim, Corrado De Vito.

**Project administration:** Floris Barnhoorn, Dineke Zeegers Paget.

**Supervision:** Corrado De Vito, Dineke Zeegers Paget, Paolo Villari.

**Visualization:** Carolina Marzuillo.

**Writing – original draft:** Annalisa Rosso, Erica Pitini.

**Writing – review & editing:** Annalisa Rosso, Erica Pitini, Elvira D'Andrea, Corrado De Vito, Paolo Villari.

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
