## [Decision Letter · Decision Letter 0]

2 Jan 2020

PONE-D-19-26230

Genomics knowledge and attitudes among European public health professionals: results of a cross-sectional survey.

PLOS ONE

Dear Dr. Rosso,

Thank you for submitting your manuscript to PLOS ONE. After careful consideration, we feel that it has merit but does not fully meet PLOS ONE’s publication criteria as it currently stands. Therefore, we invite you to submit a revised version of the manuscript that addresses the points raised during the review process.

We would appreciate receiving your revised manuscript by Feb 16 2020 11:59PM. To enhance the reproducibility of your results, we recommend that if applicable you deposit your laboratory protocols in protocols.io, where a protocol can be assigned its own identifier (DOI) such that it can be cited independently in the future. For instructions see: http://journals.plos.org/plosone/s/submission-guidelines#loc-laboratory-protocols

We look forward to receiving your revised manuscript.

Kind regards,

Lawrence Palinkas

Academic Editor

PLOS ONE

Journal Requirements:

2. Please provide additional details regarding participant consent.

In the ethics statement in the Methods and online submission information, please ensure that you have specified (i) whether consent was informed and (ii) what type you obtained (for instance, written or verbal), and how it was recorded. 

If the need for consent was waived by the ethics committee, please include this information

Reviewers' comments:

Reviewer's Responses to Questions

**Comments to the Author**

1. Is the manuscript technically sound, and do the data support the conclusions?

Reviewer #1: Yes

Reviewer #2: Yes

2. Has the statistical analysis been performed appropriately and rigorously? 

Reviewer #1: Yes

Reviewer #2: Yes

3. Have the authors made all data underlying the findings in their manuscript fully available?

Reviewer #1: Yes

Reviewer #2: Yes

4. Is the manuscript presented in an intelligible fashion and written in standard English?

Reviewer #1: Yes

Reviewer #2: Yes

5. Review Comments to the Author

Reviewer #1: The paper provides a very interesting view on knowledge and attitudes of a sample of European public health professionals on genomics and personalized medicine. The topic is of major public health interest and the paper deserves publication. However, the very low response rate (only 10% of member of the European Public Health Association have answered the survey) represents an important limitation, as also acknowledge by the authors. In this regard, they have correctly indicated that “the low response rate reflects one of the main challenges of web-based surveys”. However, this point requires some further elaboration in order to support the quality of the results obtained, adding references to published literature (e.g. confirming that the response rate obtained is in line with the average rate of web base survey?). The authors could also discuss issues of representativeness of the sample (is the EUPHA members population different in composition than the study sample?). When describing the results, information on the provenience of the respondents could be added, again to provide an idea of the actual representativeness of the sample.

There are also some other suggestions related to the description of methods, results and discussion sections.

Methods

The paper refers to a previous study where the questionnaire was piloted. However, I would suggest providing clearer definitions of the terminology adopted and explaining how were the questions designed. E.g., in section E the authors write “The following statements are based on published literature dealing with the relationship between public health and human genomics and the possible role of public health professionals in putting PHG into practice.” What is the literature they are referring to? Did they use pre-existing questions coming from similar questionnaires (eg the study by Marzuillo et al, with some of the authors also belonging to this research group))?

Even if provided in the study questionnaire annexed to the manuscript, a clear definition of what is meant by “genetic testing” should be provided by the authors in the text: are they referring to predictive testing only? Similarly, I would suggest adding a definition of “genetic service”, which is widely used throughout the text.

Results/discussion:

As also stated before, it would be worth discussing how questions, particularly those belonging to section E of the questionnaire, were designed. I believe that there may be a social desirability bias in the responses provided, and I would suggest adding this point to the discussion.

Reviewer #2: The manuscript addresses an interesting question, which is to what extent do public health professionals have competence in genomics. The investigators obtained a sample of over 600 members of the European Public Health Association to identify both level of knowledge and attitudes regarding genetic testing and genetic services. The study is largely descriptive, with logistic regression analyses conducted to assess independent predictors of knowledge and attitudes.

The finding that only 29% of participants correctly identified all medical conditions for which there is (or not) evidence for implementing genetic testing and over 60% thought that investing in genomics may divert economic resources from social and environmental determinants of health are in themselves quite noteworthy and worthy of publication. However, the manuscript leaves three important questions unanswered. First, it provides no recommendations on how adequate knowledge and capacity among PH professionals can be achieved to facilitate the integration of genomic information into PH activities. Second, given the generally positive attitudes of most PH professionals towards genetic testing and genetic services, how much knowledge within the field is adequate? Third, the significance of the finding that 60% of the sample believed it was more important to invest resources in the social and economic causes of ill health than in the implementation of genetic testing does not mean that they believe genetic testing is unimportant. I believe a more nuanced interpretation of the finding must be provided in the discussion given the wording of the question and the implications of this finding for integrating genomics into public health. The authors need to explain why the belief that genomics are important but not as important as social and environmental determinants constitutes a barrier to their implementation in public health.

As the authors indicate, the response rate is very low, which raises questions regarding the generalizability of the findings and potential for response bias. The authors note on lines 143-145 that there were no significant socio-demographic differences between those who completed the entire survey and those who did not; however, that is not the same as comparing the socio-demographic differences between those who agreed to participate and those who did not. Given that one of the authors is from the European Public Health Association, surely there must be some demographic information of the membership of the association as a whole (19,000 according to the association website) that can be used to assess how representative of that membership.

It would also be helpful to provide information on the nationality of the study participants and a comparison of knowledge and attitudes by nationality. The authors note on lines 274-277 that organizational models of genetic services may differ in different countries. However, this was not included as a covariate in the regression analyses. However, a table of comparisons of knowledge and attitudes by nationality as well as other socio-demographic characteristics would be helpful.

In the multivariate analysis, knowledge is predictive of positive attitudes towards genetic testing and delivery of genetic services, but not of positive attitudes towards use of resources for genetic testing or positive attitudes toward the role of professionals in PHG. The significance of this finding should be addressed in the discussion, especially as it raises the question of what can be done to change attitudes without having to rely on increased education alone. The findings would seem to contradict the recommendation that greater awareness is needed.

6. PLOS authors have the option to publish the peer review history of their article (what does this mean?). If published, this will include your full peer review and any attached files.

Reviewer #1: No

Reviewer #2: No

---

## [Author Response · Author response to Decision Letter 0]

29 Feb 2020

Dear Editor and Reviewers,

Apologizing for the delay in providing our feedback, also due to work overload caused by the current SARS-COV2 outbreak in Italy, we would like to thank the reviewers for addressing several important points in the manuscript with their comments. We tried to include all their suggestions to improve the quality of the paper, expanding both the methods, results and discussion sections. We have also added ten additional references, including reference to the recommendations of PRECeDI project, which were released very recently. Finally, we added a sentence regarding the request for consent and the approval received by Sapienza University Ethics Committee to conduct the study in the methods section.

Please find below our responses to all comments of the two reviewers.

Reviewer #1

1. The Reviewer wrote “The very low response rate (only 10% of member of the European Public

Health Association have answered the survey) represents an important limitation, as also acknowledge by the authors. In this regard, they have correctly indicated that “the low response rate reflects one of the main challenges of web-based surveys”. However, this point requires some further elaboration in order to support the quality of the results obtained, adding references to published literature (e.g. confirming that the response rate obtained is in line with the average rate of web base survey?). The authors could also discuss issues of representativeness of the sample (is the EUPHA members population different in composition than the study sample?). When describing the results, information on the provenience of the respondents could be added, again to provide an idea of the actual representativeness of the sample.”

We agree with the Reviewer that the low response rate represents an important limitation in our study and that it should be better addressed in the Discussion section. We have added some references to published evidence on reported response rates, in order showing that “The response rate obtained in our study is coherent with that of other studies that also relied on web-based surveys. Based on the results of different studies, Ban Mol reported in fact that a response rate below 10% is not uncommon for web surveys (which have been shown to generally get a 6 to 15% lower response rate compared to other survey modes (Ban Mol C. Improving web survey efficiency: the impact of an extra reminder and reminder content on web survey response, International Journal of Social Research Methodology. 2017 20:4:317-327, DOI: 10.1080/13645579.2016.1185255),).” We have also specified that “we only sent a limited number of remainders (three emails were sent in total to EUPHA members), considering that repeated follow-ups have been shown to diminish returns and may have the counterproductive effect to irritate potential respondents, without noticeably increasing response rates (Deutskens, E. C., Ruyter, de, J. C., Wetzels, M. G. M., & Oosterveld, P.Response rate and response quality of internet-based surveys: an experimental study. Marketing Letters. 2004,15(1): 21-36. https://doi.org/10.1023/B%3AMARK.0000021968.86465.00).

We have also added a sentence discussing representativeness of the sample, writing “Despite the low response rate obtained, the sample proves to be representative of the population of EUPHA members in terms of professional background (a recent survey conducted by the EUPHA secretariat indicated that 66% of respon s identified themselves as researchers, 14% as Policymakers and the remaining 20% as Practitioners (data not published), showing a similar distribution than our sample)”. 

We thank for the suggestion to also include information on the nationality of respondents in the Results section. We have added the following paragraph “Respondents came from all EU28 Countries and some non –EU Countries, including Albania (n=1), Andorra (n=1), Bosnia and Herzegovina (n=4), Norway (n=15), Russia (n=1), Serbia (n=3), Switzerland (n=25), Turkey (n=7), and there were 27 people working in developing countries at the time the survey was conducted”

2. The Reviewer wrote “The paper refers to a previous study where the questionnaire was piloted. However, I would suggest providing clearer definitions of the terminology adopted and explaining how were the questions designed. E.g., in section E the authors write “The following statements are based on published literature dealing with the relationship between public health and human genomics and the possible role of public health professionals in putting PHG into practice”. What is the literature they are referring to? Did they use pre-existing questions coming from similar questionnaires (eg the study by Marzuillo et al, with some of the authors also belonging to this research group))?”

We agree with the Reviewer that we did not clarify enough how the questionnaire was developed. Our manuscript included a reference on the paper presenting the results of the pilot phase of the survey, where more information on the questionnaire are provided. However, we have also added a sentence clarifying that “The development of the questionnaire was informed by a literature review and by similar studies previously carried out by the same research group (18,22). It was also based on the inputs received by the members of EUPHA Section on PHG and by all partners of PRECeDI project, as described elsewhere (23).”

3. The Reviewer wrote “Even if provided in the study questionnaire annexed to the manuscript, a clear definition of what is meant by “genetic testing” should be provided by the authors in the text: are they referring to predictive testing only? Similarly, I would suggest adding a definition of “genetic service”, which is widely used throughout the text.”

We agree with the Reviewer that a definition of the terms “genetic testing” and “genetic service” should have been provided also in the manuscript. We have added the following paragraph in the methods section:

“For the purpose of the survey, we defined genetic testing as “performing a type of medical test involving an analysis of human chromosomes, DNA, RNA, genes, and/or gene products (e.g., enzymes and other types of proteins), which is predominately used to detect heritable or somatic mutations, genotypes, or phenotypes related to disease and health”, following the definition proposed by Burke (24). Further to discussion with other PRECEDI project partners, we have decided to adopt the classification of genetic testing used by the European Society of Human Genetics (25). In particular, the survey was aimed at investigating knowledge and attitudes towards the implementation of susceptibility (also known as predisposition) tests, defined as “the detection of genetic variants that are associated with an increased risk of disease but cannot predict with certainty the development of disease, because of the incomplete penetrance of the genetic mutation”. Examples of this type of tests include BRCA testing, Lynch syndrome testing and familial hypercholesterolemia genetics tests. A “genetic service” was defined as “the provisions to diagnose, advise and treat individuals with risk factors for genetic disorders.” According to our interpretation, the delivery of a genetic service is therefore expected to provide not only genetic testing and counselling, but also treatment and follow up of individuals with genetic disorders, similarly to what was previously described by Battista et al.”( 26)

4. The Reviewer wrote “As also stated before, it would be worth discussing how questions, particularly those belonging to section E of the questionnaire, were designed. I believe that there may be a social desirability bias in the responses provided, and I would suggest adding this point to the discussion.”

We thank the reviewer for this comment. We had described the questionnaire development in the paper presenting the results of the pilot phase of the survey, but we agree that it would be useful to provide more information on the process also in this paper. We have included a sentence clarifying that “Statements included in section E were based on published literature dealing with the relationship between public health and human genomics and the possible role of public health professionals in putting PHG into practice (27)”.

Reviewer #2

1. The Reviewer wrote “The finding that only 29% of participants correctly identified all medical conditions for which there is (or not) evidence for implementing genetic testing and over 60% thought that investing in genomics may divert economic resources from social and environmental determinants of health are in themselves quite noteworthy and worthy of publication. However, the manuscript leaves three important questions unanswered. First, it provides no recommendations on how adequate knowledge and capacity among PH professionals can be achieved to facilitate the integration of genomic information into PH activities. Second, given the generally positive attitudes of most PH professionals towards genetic testing and genetic services, how much knowledge within the field is adequate? Third, the significance of the finding that 60% of the sample believed it was more important to invest resources in the social and economic causes of ill health than in the implementation of genetic testing does not mean that they believe genetic testing is unimportant. I believe a more nuanced interpretation of the finding must be provided in the discussion given the wording of the question and the implications of this finding for integrating genomics into public health. The authors need to explain why the belief that genomics are important but not as important as social and environmental determinants constitutes a barrier to their implementation in public health.”

We thank the reviewer for this comprehensive comment, which allowed us to address interesting issues in the Discussion section of the manuscript. 

With regards to the way in which knowledge and capacity among PH professionals can be achieved to facilitate the integration of genomic information into PH activities, we have elaborated more in the discussion section on the need to increase the genomic competences of PH professionals in Europe. “Healthcare professional education should be the first step to increase knowledge and capacity of professionals on genomics. There are significant differences in the way in which professional education is delivered across the countries of Europe, and also the study curricula of physicians and non physician PH specialists may differ substantially. A set of core competencies in genomics for health professionals (even though not specifically addressed at PH professionals) has been proposed by the European Society of Human Genetics, to provide an appropriate framework for establishing minimum standards of preparation and guide the development of study curricula for health-care professionals in all Countries (graduate or post- graduate) (45). Based on this framework, a study conducted in Italy tried to identify a set of core competencies in genetics for non-geneticists, both physicians and non-physicians, developing a proposal of three different curricula according to the profession, including basic knowledge, but also attitudes and abilities needed to be able to effectively incorporate genomics into practice (46).

We believe that our previous response could also answer to the Reviewer’s question on how much knowledge within the genetic field is adequate. Overall, there is scarce evidence on PH practitioners’ capacity in this area, ours was one of the few studies conducted on this specific topic, and no study so far has ever tried to estimate a miminum level of knowledge required to properly include genomic applications into PH practice. However, as already stated, core competencies needed to properly incorporate genetics into practices have been proposed by different scientific societies both at the European and national level. 

 Finally, with regards to the third point raised by the reviewer, we agree that those answering that “investing resources in the social and economic causes of ill health is more important than investing them in the implementation of genetic testing” do not necessary believe that genetic testing is unimportant. However, in a time where public health budgets are suffering, they may fear that investing in public health genomics and precision medicine could divert resources from addressing the “traditional” determinants of health, as already stated in the discussion. We have added the following sentence in the Discussion “In this regard, there is a long-standing debate in the PH community on the opportunity to focus on -omics and personalized medicine, and whether this may divert from the traditional population-based approach. The need to go beyond the dichotomous high-risk versus population approach was suggested, taking into account that precision public health could contribute to improving population health and achieving social justice—equity, social inclusion, and empowerment, through the use of individual data.”

2. The Reviewer wrote “As the authors indicate, the response rate is very low, which raises questions regarding the generalizability of the findings and potential for response bias. The authors note on lines 143-145 that there were no significant socio-demographic differences between those who completed the entire survey and those who did not; however, that is not the same as comparing the socio-demographic differences between those who agreed to participate and those who did not. Given that one of the authors is from the European Public Health Association, surely there must be some demographic information of the membership of the association as a whole (19,000 according to the association website) that can be used to assess how representative of that membership.”

We agree with the Reviewer that some more information on the generalizability of the results would be needed. Unfortunately, the EUPHA secretariat does not have detailed information on the socio-demographic characteristics of its of the Association members. In this regard, we would like to clarify that the reference population for your survey was that of EUPHA members, which account to approximately 5,000 people, much smaller than the 19,000 reported in the website, which refer to the subscribers to the EUPHA Newsletter (the so called “EUPHA Network”). EUPHA is an umbrella organization for public health associations and institutes in Europe, therefore its members are all individual members of national public health associations, who, when subscribing to their national association, have indicated to be interested in European public health affairs and policies and paid an additional fee, which entitles them to a subscription to the European Journal of Public Health and reduced fees for the EPH Conference. The only information provided by national public health associations to EUPHA secretariat is their members’ e-mail address. Early 2019, the Secretariat did a very short web survey on the professional background of EUPHA Newsletter subscribers. Results, based on 170 responses only, indicated that 66% of responders identified themselves as researchers, 14% as Policymakers and the remaining 20% as Practitioners. These results are similar to those reported in our survey, with 58.8% of respondents defined as Academics (researchers), 19.7% working at the Governmental level, and the remaining working in different types of practices (Hospital, Local Health Units, NGOs, etc.). This information was included in the manuscript.

3. The Reviewer wrote “It would also be helpful to provide information on the nationality of the study participants and a comparison of knowledge and attitudes by nationality. The authors note on lines 274-277 that organizational models of genetic services may differ in different countries. However, this was not included as a covariate in the regression analyses. However, a table of comparisons of knowledge and attitudes by nationality as well as other socio-demographic characteristics would be helpful.”

We thank the Reviewer for this comment. We agree that it would have been interesting to compare differences in attitudes and knowledge by nationality, however, responses came from a too high number of Countries, not allowing to include it as a covariate in the analysis (the number of observation for each category would have been to small). We could have combined countries according to the type of organization of genetic services, but unfortunately, information on the organization of delivery models in European countries is not available: within PRECeDI project, we attempted to conduct an expert survey in Europe to assess the delivery of genetic services and adherence to international guidelines for genetic testing for BRCA, Lynch Syndrome, Familial Hypercholesterolemia and Hereditary Trombophilia, but we were able to collect responses only from a limited number of Countries (six to twelve Countries, according to the type of hereditary condition assessed) (Rosso A, et al. European Journal of Public Health, Volume 28, Issue suppl_4, November 2018, cky213.371,) We have added the following sentence in the Discussion section of the paper, when addressing the limitations of the study “Another limitation of the study was the impossibility to compare knowledge and attitudes on genomics across different Countries: respondents came from a high number of different counties (over 35), not allowing to stratify responses according to the provenience. The lack of information on the organizational models of genetic services in all Countries did not allow to control for type of delivery model either (50).” However, as previously reported in response to Reviewer #1, we have included information on the nationality of study participants.

4. In the multivariate analysis, knowledge is predictive of positive attitudes towards genetic testing and delivery of genetic services, but not of positive attitudes towards use of resources for genetic testing or positive attitudes toward the role of professionals in PHG. The significance of this finding should be addressed in the discussion, especially as it raises the question of what can be done to change attitudes without having to rely on increased education alone. The findings would seem to contradict the recommendation that greater awareness is needed.

We thank the reviewer for this comment, which suggested us that some information should also be provided on the results of the univariate analysis. Actually, both the association between knowledge and positive attitudes towards the use of resources for genetic testing and with positive attitudes toward the role of professionals in PHG were statistically significant at the univariate analysis P=0.016 annd P=0.021). However, when controlling for other covariates, knowledge appeared to be less relevant than other factors, mainly the fact of having PHG as one of the main areas of work (which, in turn, is one of the strongest predictors of knowledge). We have added a paragraph explaining the association between knowledge and attituded in the Discussion, and we also added a sentence saying that “Given the association observed between a direct involvement in PHG activities and both knowledge and attitudes, it may be useful to develop specific training initiatives for the PH workforce in framework of continuous medical education and/or on-job training”. 

We hope that the manuscript can now be considered suitable for publication.

Sincerely,

Annalisa Rosso and all co-authors

---

## [Editor Report · Decision Letter 1]

9 Mar 2020

Genomics knowledge and attitudes among European public health professionals: results of a cross-sectional survey.

PONE-D-19-26230R1

Dear Dr. Rosso,

We are pleased to inform you that your manuscript has been judged scientifically suitable for publication and will be formally accepted for publication once it complies with all outstanding technical requirements.

With kind regards,

Lawrence Palinkas

Academic Editor

PLOS ONE
---

## [Editor Report · Acceptance letter]

18 Mar 2020

PONE-D-19-26230R1 

Genomics knowledge and attitudes among European public health professionals: results of a cross-sectional survey. 

Dear Dr. Rosso:

I am pleased to inform you that your manuscript has been deemed suitable for publication in PLOS ONE. Congratulations! Your manuscript is now with our production department. 

With kind regards,

on behalf of

Dr. Lawrence Palinkas 

Academic Editor

PLOS ONE